# A spider silk-derived solubility domain inhibits nuclear and cytosolic protein aggregation in human cells

Anna Katharina Schellhaus[1], Shanshan Xu[1], Maria E. Gierisch[1], Julia Vornberger[1], Jan Johansson[2] & Nico P. Dantuma [1✉]

Due to the inherent toxicity of protein aggregates, the propensity of natural, functional amyloidogenic proteins to aggregate must be tightly controlled to avoid negative consequences on cellular viability. The importance of controlled aggregation in biological processes is illustrated by spidroins, which are functional amyloidogenic proteins that form the basis for spider silk. Premature aggregation of spidroins is prevented by the N-terminal NT domain. Here we explored the potential of the engineered, spidroin-based NT* domain in preventing protein aggregation in the intracellular environment of human cells. We show that the NT* domain increases the soluble pool of a reporter protein carrying a ligand-regulatable aggregation domain. Interestingly, the NT* domain prevents the formation of aggregates independent of its position in the aggregation-prone protein. The ability of the NT* domain to inhibit ligand-regulated aggregation was evident both in the cytosolic and nuclear compartments, which are both highly relevant for human disorders linked to non-physiological protein aggregation. We conclude that the spidroin-derived NT* domain has a generic anti-aggregation activity, independent of position or subcellular location, that is also active in human cells and propose that the NT* domain can potentially be exploited in controlling protein aggregation of disease-associated proteins.

[1] Department of Cell and Molecular Biology, Karolinska Institutet, Solnavägen 9, S-17165 Stockholm, Sweden. [2] Department of Biosciences and Nutrition, Karolinska Institutet, Neo, S-14183 Huddinge, Sweden. ✉email: nico.dantuma@ki.se

The plethora of human diseases that are linked to proteins that precipitate into insoluble aggregates highlights the importance for cells to keep their intracellular environment free of these inherently toxic protein species[1]. The devastating effects of aggregation-prone proteins are clearly illustrated by neurodegenerative diseases, such as Alzheimer's, Parkinson's and Huntington's disease, which are characterized by an age-dependent accumulation of insoluble protein aggregates in affected neurons or brain parenchyma[2]. Although neurological disorders may be overrepresented, probably due to the limited possibilities of long-lived, post-mitotic neurons to eliminate protein aggregates, protein misfolding disorders are by no means limited to neurodegeneration, but also include diseases in which other tissues are affected by aggregation-prone proteins[3]. Even though the exact nature of the toxic species is still subject to debate, there is a large body of evidence suggesting that the propensity of proteins to aggregate is the primary determinant for the toxicity they elicit in these diseases[4]. Accordingly, keeping the disease-associated proteins in a soluble state is expected to minimize their cytotoxic effects and is, as such, a desirable but challenging objective from a therapeutical perspective.

The strength, robustness and hard-to-disentangle nature of protein aggregates are at the same time unique properties that endow aggregation-prone proteins with the ability to form exceptionally rigid structures that can deal with forces that would be difficult, if not impossible, to resist by natively folded, globular proteins[5]. This exquisite characteristic explains the presence of amyloidogenic domains in a variety of natural proteins for which the unique features of these domains outweigh the costs of keeping the potential toxicity of these proteins at bay. A prime example is the family of amyloidogenic proteins called spidroins, which form the amyloid-like fibers that are the main constituent of spider silk, giving this biological product its unique properties and exceptional strength[6,7].

Spidroins are present at very high concentrations in the spider's gland but only form safely amyloid-like fibers when secreted, which demands a stringent and tightly controlled mechanism to prevent premature aggregation in the gland. This is accomplished by the N-terminal (NT) domain of spidroins that precedes the highly repetitive amyloidogenic domain and regulates the formation of amyloid fibrils in a pH- and $CO_2$-dependent fashion[8]. The NT domain has a dual function as it not only prevents premature aggregation but also, when appropriate, promotes aggregation by forming NT-dimers that promote oligomerization of the amyloidogenic domain[9]. An engineered NT domain, referred to as NT*, has been optimized to prevent protein aggregation. The NT* contains two amino acid substitutions, (Asp 40 to Lys, Lys 65 to Asp), which result in loss of the dipolar nature of wild-type NT that is required for domain-domain charge attraction[10,11]. Consequently, the NT* domain does not display the pH-dependent dimerization in vitro, while still preventing protein aggregation[12]. The NT* domain has been explored in various experimental models for its ability to prevent aggregation of proteins-of-interest[13–18]. The spidrion-derived NT* anti-aggregation domain has also been found to efficiently prevent aggregation of bacterially produced recombinant proteins, underscoring its usefulness as a solubility tag[12,14]. Interestingly, the ability of the NT domain to prevent protein aggregation appears to be its primary, native function, unlike other commonly used solubility tags, such as glutathione S transferase (GST), maltose binding protein (MBP), and ubiquitin[19], which motivated us to examine its anti-aggregation potential in the natural environment where disease-associated amyloidogenic proteins have devastating effects.

In this study, we explored the anti-aggregation properties of the NT* domain in the cytosolic and nuclear compartment of human cells using the aggregation destabilization domain (AgDD)[20]. The AgDD domain, which is derived from the human FK506 binding protein 12 (FKBP12) protein[21], has the unique feature that its aggregation state can be regulated through administration of the stabilizing ligand Shield-1[20]. Using this experimental model, we found that the spider silk-derived NT* domain can efficiently prevent protein aggregation in cellular compartments relevant for protein misfolding diseases. This proof-of-principle study suggests that the generic anti-aggregation domain of a natural spider protein may be exploitable to counteract the toxic properties of disease-associated, aggregation-prone proteins.

## Results

**Spider NT* domain prevents ligand-regulatable aggregation in human cells.** Our aim was to test if the anti-aggregation properties of the NT* domain can be exploited in preventing protein aggregation in human cells. For our experiments, we selected the ligand-regulatable aggregation domain AgDD. The AgDD domain is a modified version of a destabilization domain (DD) that targets proteins for degradation in the absence of the ligand Shield-1[22]. By adding an additional 10 amino-acid sequence to the N-terminus of the DD signal, this motif has been converted to a Shield-1-regulatable aggregation domain that allows rapid induction of protein aggregation by omission of the ligand[20]. We chose this engineered aggregation domain because of its regulatable nature and its ability to promote the formation of filamentous protein aggregates that resemble the aggregates observed with disease-associated proteins[20]. The finding that AgDD induces protein aggregation irrespective of its position in the host protein also allowed a modular approach. For easy detection and to avoid interference of protein aggregation on the folding and maturation of the fluorescent protein, we combined these domains with superfolder green fluorescent protein (sfGFP)[23]. Thus, the reporter proteins were composed of three modules: the AgDD aggregation domain, the NT* anti-aggregation domain, and sfGFP (Fig. 1a).

In the first set of experiments, we analyzed if the NT* domain could reduce aggregation of a fusion protein carrying an N-terminal AgDD domain as it has been shown to induce aggregation most efficiently in this position[20]. The fusion proteins were transiently expressed in HeLa cells and analyzed by western blotting for protein aggregation. As expected, AgDD-sfGFP displayed Shield-1-regulatable aggregation as omission of Shield-1 resulted in a significant decrease in the pool of soluble AgDD-sfGFP (Fig. 1b, c). Importantly, insertion of the NT* anti-aggregation domain in between the AgDD and sfGFP modules resulted in an increase in soluble fusion protein in the absence of Shield-1, suggesting that protein aggregation was efficiently prevented by insertion of the NT* domain (Fig. 1b, c).

**Anti-aggregation effect of NT* is independent of its position within the host protein.** To test the impact of the position of the NT* domain, we compared the ability of NT* to prevent aggregation when placed at the amino terminus, or with a centrally, as above, or carboxy terminally positioned NT* domain (Fig. 2a). HeLa cell lines stably expressing NT*-AgDD-sfGFP, AgDD-NT*-sfGFP or AgDD-sfGFP-NT* were generated as well as a control HeLa cell line that expressed the AgDD-sfGFP reporter lacking the NT* domain. Flow cytometry showed that the stable cell lines expressing the fusions with the internal and C-terminal NT* had comparable steady-state levels of the fusion proteins as cells expressing AgDD-sfGFP whereas the fusion with the N-terminal NT* domain was expressed at slightly elevated levels (Fig. 2b). This result agrees with the recent finding that

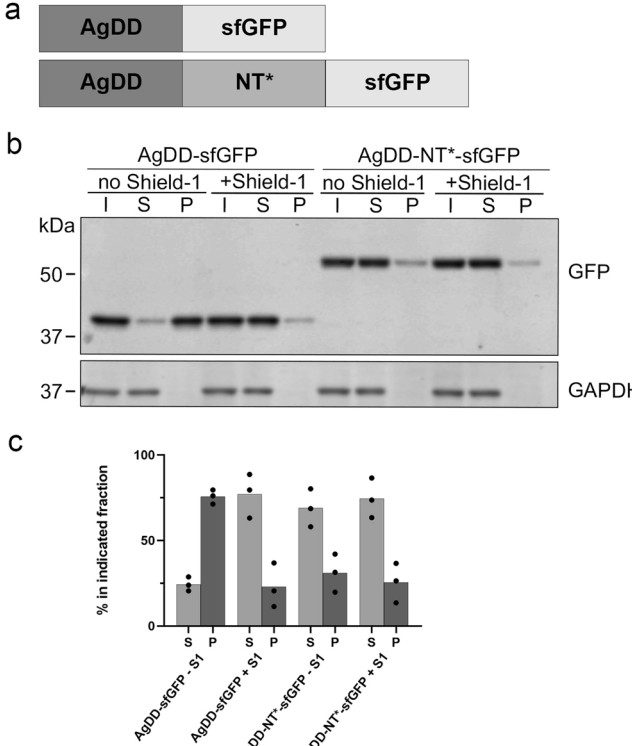

**Fig. 1 Spider NT\* domain prevents ligand-regulatable aggregation in human cells. a** Schematic representation of the fusion protein AgDD-sfGFP and AgDD-NT\*-sfGFP. **b** HeLa cells were transiently transfected with AgDD-sfGFP or AgDD-NT\*-sfGFP, incubated in the absence or presence of 1 μM Shield-1 and fractionated into Triton X-100 soluble (S) and Triton X-100 insoluble/SDS soluble (P) fractions. (I) shows the whole cell lysate/input before fractionation. Western blots were probed with antibodies directed against GFP and GAPDH. **c** Quantification of the relative distribution of the GFP band in the soluble and insoluble fractions.

N-terminally located NT\* can increase the translational efficiency of target proteins[24]. In the AgDD-sfGFP-expressing cells, the majority of the fusion protein was detergent insoluble in the absence of Shield-1 (Fig. 2c). Western blot analysis confirmed the anti-aggregation effect of the internally positioned NT\* domain in the stable cell line as no detectable signal was obtained in the detergent insoluble material with antibodies directed against GFP and NT\* even in the absence of Shield-1 (Fig. 2c). This effect was indistinguishable from the effects observed with the amino and carboxy terminal NT\* domains, suggesting that its ability to prevent aggregation is not affected by the position of the NT\* domainn in the host protein (Fig. 2c). Administration of Shield-1 rendered the NT\*-lacking AgDD-sfGFP comparably soluble as the effect of insertion of the NT\* domain (Fig. 2d). Microscopic analysis revealed primarily cytosolic aggregates in AgDD-sfGFP-expressing cells in the absence of Shield-1 but a predominant homogenous cytosolic staining of the fusions that contained the NT\* domain, consistent with its anti-aggregation properties (Fig. 2e). Addition of Shield1 to the culture medium did not only prevent the formation of the cytosolic AgDD-sfGFP aggregates but also caused a uniform redistribution of the AgDD-sfGFP in the cytosolic and nuclear compartments (Fig. 2f). Whereas the NT\*-containing fusions displayed a homogenous distribution in the absence and presence of Shield-1, we observed also here a redistribution from a predominant cytosolic localization to a uniform distribution throughout the cytosolic and nuclear compartment upon administration of Shield-1 (Fig. 2f).

As this redistribution was observed for both the control AgDD-sfGFP and the NT\*-containing fusions, this is likely a general effect of Shield-1 on the AgDD domain and hence a feature independent of the NT\* domain. This is further supported by our observation that an NT\*-sfGFP fusion displays a uniform distribution throughout the cytosolic and nuclear compartments, showing that the NT\* domain does not affect the localization (Supplementary Fig. 1).

**NT\* inhibits ligand-regulatable aggregation in the nucleus.** The presence of nuclear inclusions is a common phenomenon in neurodegenerative diseases[25]. Moreover, nuclear localization of aggregation-prone proteins has been found to be critical for their pathologic effect in several neurodegenerative diseases[26–29]. As the AgDD-sfGFP primarily caused cytosolic aggregates, the question remained whether the NT\* domain can also prevent protein aggregation in the nuclear compartment. To address this question, we provided the AgDD-sfGFP and AgDD-NT\*-sfGFP with a nuclear localization signal (NLS) and transiently expressed these proteins in HeLa cells. The NLS-AgDD-sfGFP fusion precipitated in large intranuclear aggregates in the absence of Shield-1 (Fig. 3a). Administration of Shield-1 resulted in the disappearance of the nuclear puncta and a more homogenous nuclear distribution of the fusion protein, which was modestly enriched in nucleoli. The NT\* containing fusion did not give rise to large intranuclear inclusions in the absence of Shield-1 but displayed a similar, more uniform nuclear staining in the absence or presence of Shield-1 (Fig. 3a, b). The nuclear distribution with the NLS-AgDD-NT\*-sfGFP had a granular appearance independent of the absence or presence of Shield-1. Western blotting confirmed that in the presence of Shield-1 both reporter proteins resided in the soluble fraction (Fig. 3c, d). Withdrawal of Shield1 changed the status of the vast majority of NLS-AgDD-sfGFP to detergent insoluble but this was significantly reduced by the introduction of the NT\* domain, which kept the majority of the fusion protein in a detergent soluble state in the absence of Shield1 (Fig. 3c, d). A fraction of insoluble nuclear AgDD-NT\*-sfGFP protein remained which was, however, largely unaffected by administration of Shield-1 (Fig. 3c, d). We conclude that the NT\* domain can inhibit the Shield-1-regulatable nuclear aggregation but at the same time endows the reporter protein with properties that causes a granular localization in the nuclear compartment and prevents complete solubilization.

**Effect of NT\* domain on AgDD aggregation in DAOY cells.** As various cellular processes are differently adapted in neurons[30], we tested the effect of the NT\* domain on protein aggregation in DAOY cells, a neuroectodermal cell line derived from a pediatric medullablastoma[31]. In DAOY cells, inclusion of the NT\* domain prevented aggregation of the AgDD-sfGFP fusion independent of the position of the NT\* domain in the fusion (Fig. 4a). The nuclear localized AgDD-NLS-sfGFP fusion gave rise to the formation of only one or a few nuclear foci in the absence of Shield-1 instead of the large number of nuclear aggregates that were observed in HeLa cells (Fig. 4b). Consistent with the foci being due to nuclear aggregation of the AgDD-sfGFP, administration of Shield-1 resulted in the disappearance of the nuclear foci, which was accompanied with the fusion being distributed throughout the nucleus (Fig. 4b). In DAOY cells, similar to HeLa cells, AgDD-sfGFP levels were elevated in nucleoli in the presence of ligand. The nuclear-localized fusion that contained the NT\* domain did not form nuclear foci but instead distributed equally in the nucleoplasm without being enriched in nucleoli (Fig. 4b). These data show that the NT\* domain can prevent aggregation of AgDD-containing constructs independent of its positionn in the

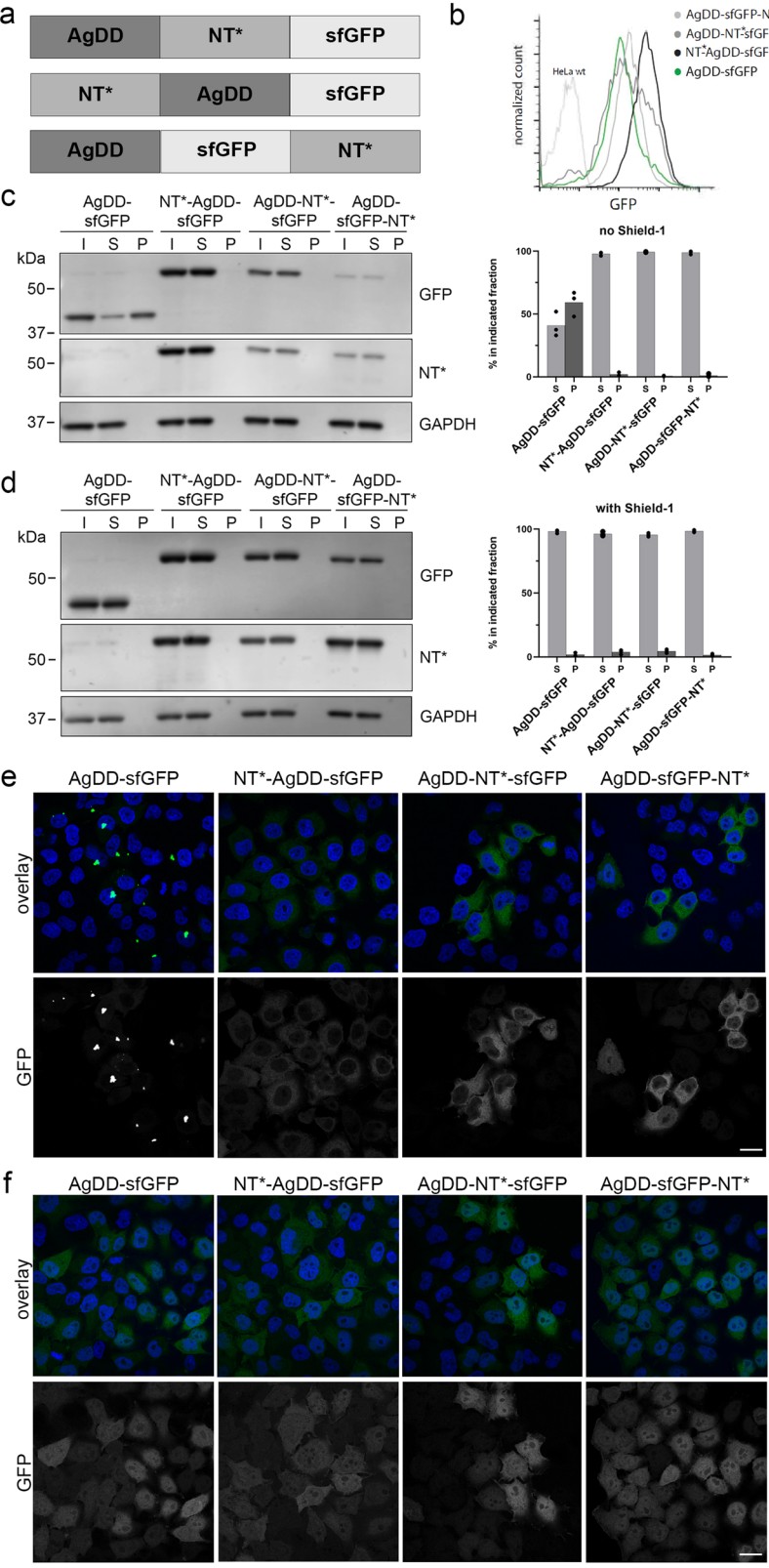

**Fig. 2 Anti-aggregation effect of NT\* is independent of its position within the host protein. a** Schematic representation of the fusion proteins with the NT\* domain in different positions. **b** HeLa cells were stably transfected with the fusion proteins depicted in (A) and analysed by flow cytometry. **c**, **d** HeLa cells stably expressing AgDD-sfGFP, NT\*-AgDD-sfGFP, AgDD-NT\*-sfGFP or AgDD-sfGFP-NT\* were incubated for 16–24 h in the absence (**c**) or presence of 1 μM Shield-1 (**d**) and fractionated into Triton X-100 soluble (S) and Triton X-100 insoluble/SDS soluble (P) fractions. Input of whole cell lysates before fractionation are shown (I). Western blots were performed with antibodies directed against GFP, the NT\*-tag and GAPDH. Quantifications of the relative distribution of the GFP band in the soluble and insoluble fractions from three independent experiments are shown (**e**, **f**). HeLa cells stably expressing AgDD-sfGFP, NT\*-AgDD-sfGFP, AgDD-NT\*-sfGFP or AgDD-sfGFP-NT\* in the absence (**e**) or presence (**f**) of 1 μM Shield-1 were fixed in 4% PFA and counterstained with Hoechst. Scale bar is 20 μm.

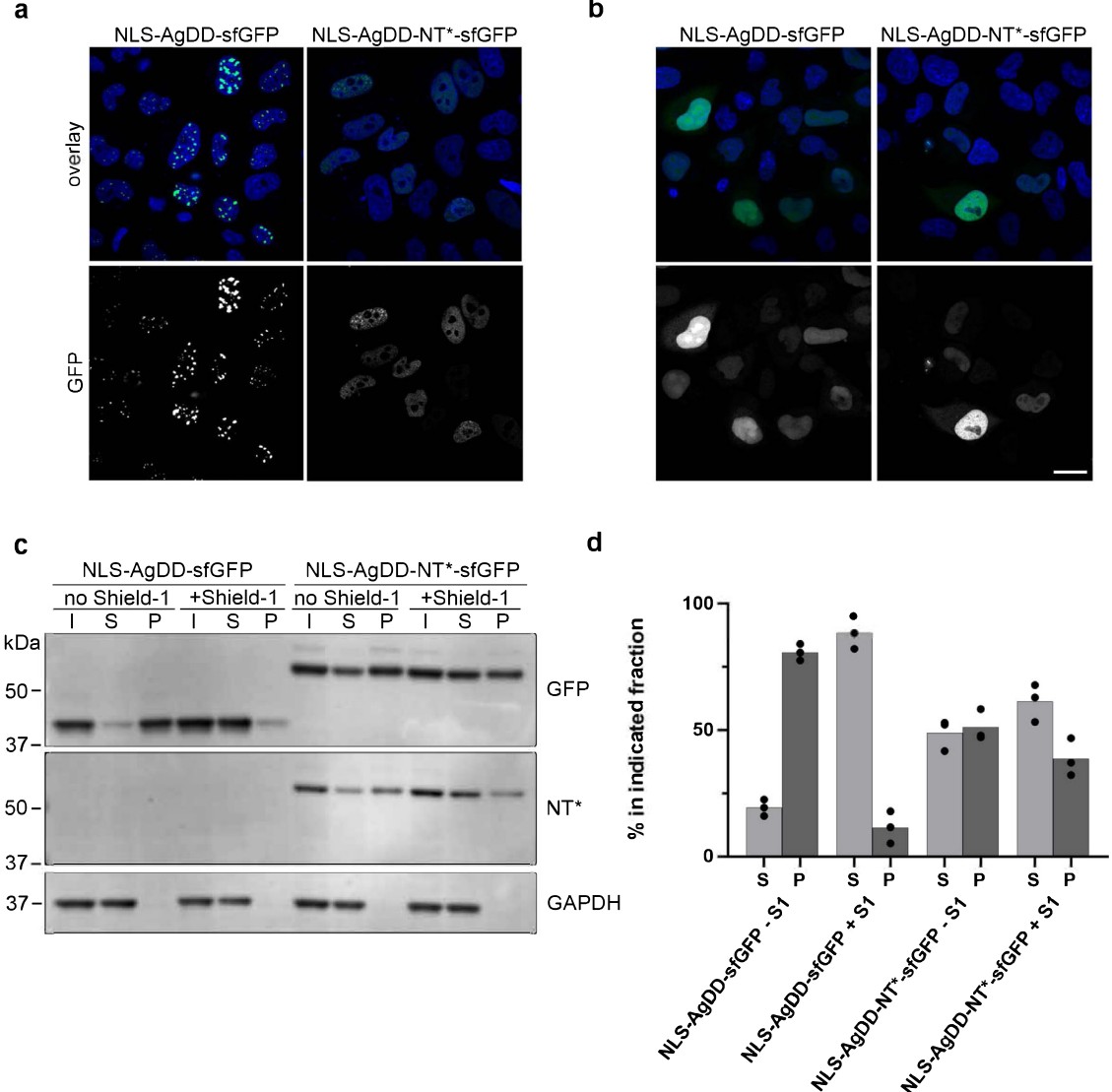

**Fig. 3 NT\* inhibits ligand-regulatable aggregation in the nucleus. a, b** HeLa cells were transiently transfected with NLS-AgDD-sfGFP or NLS-AgDD-NT\*-sfGFP and incubated in the absence (**a**) or presence (**b**) of 1 μM Shield-1. Cells were fixed 24 h after transfection with 4% PFA and counterstained with Hoechst. Scale bar is 20 μm. **c** HeLa cells transiently transfected with NLS-AgDD-sfGFP or NLS-AgDD-NT\*-sfGFP were incubated for 16–24 h in the absence of presence of 1 μM Shield-1 and fractionated into Triton X-100 soluble (S) and Triton X-100 insoluble/SDS soluble (P) fractions. Input of whole cell lysates before fractionation are shown (I) Western blots were performed with antibodies directed against GFP, the NT\*-tag and GAPDH. **d** Quantifications of the relative distribution of the GFP band in the soluble and insoluble fractions from three independent experiments are shown.

fusion protein and is active both in the cytosolic and nuclear compartment.

## Discussion

A large number of neurodegenerative diseases are characterized by the presence of deposits of protein aggregates in affected neurons. Familial forms of neurodegenerative diseases are often caused by mutant proteins that have an increased propensity to precipitate into insoluble aggregates further underscoring the tight link between protein aggregation and neurodegeneration[32]. Indeed, protein aggregates are inherently toxic to neuronal cells[4] and drugs that prevent aggregation can limit the toxicity elicited by these proteins[33]. This realization has sparked an interest in the development of therapeutic interventions that inhibit aggregation of disease-associated proteins. Development of strategies and tools that specifically and efficiently prevent the culprit proteins from forming aggregates is, however, a challenging endeavor

because of the generic nature of protein aggregation and the supersaturated protein concentrations in the intracellular environment.

Lessons learned from nature's solutions to physiological challenges can give important insights for our own attempts to modulate these processes in human diseases. The emerging understanding that proteins in an insoluble, aggregated state are not only involved in the aetiology of diseases but also of critical importance in a variety of physiological processes implies that cells must have developed their own strategies to control the potential toxicity of these amyloidogenic proteins[5]. Given the common principles of protein aggregation it is maybe not surprising that a spider-derived solubility domain can prevent aggregation of human proteins[12], in a similar way that it temporally and spatially controls the oligomerization of spidroins into spider silk[9]. In the present study, we show that this anti-aggregation property remains active in the cytosolic and nuclear

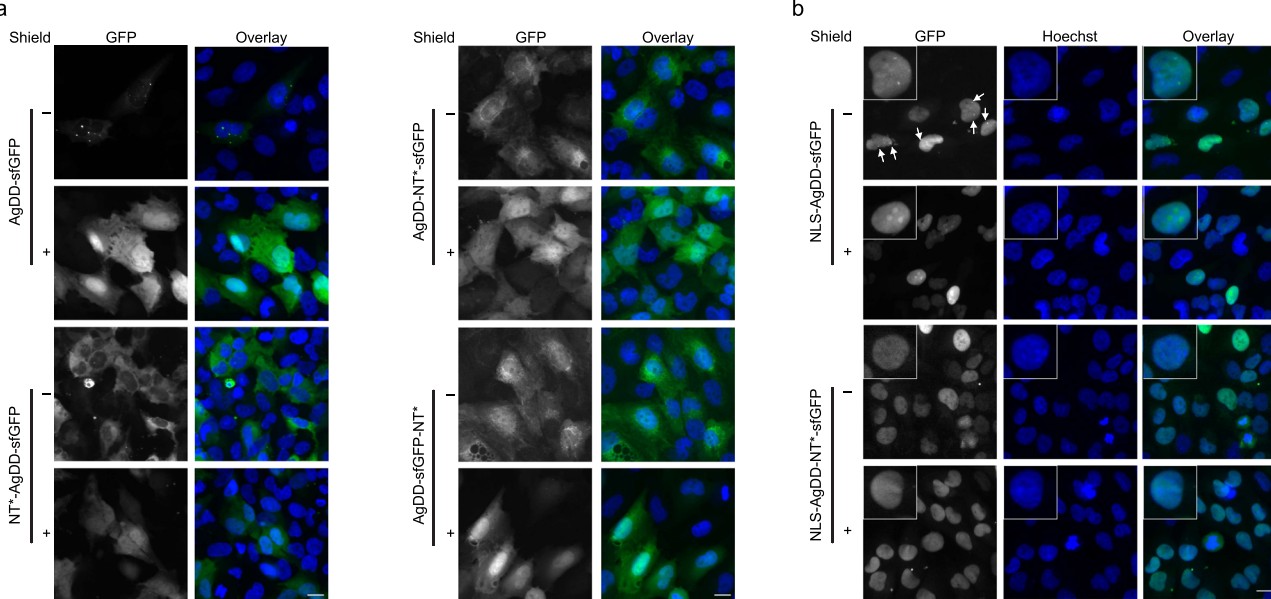

**Fig. 4 Effect of NT\* domain on AgDD aggregation in DAOY cells. a** DAOY cells stably expressing AgDD-sfGFP, NT\*-AgDD-sfGFP, AgDD-NT\*-sfGFP or AgDD-sfGFP-NT\* in the absence or presence of 1 μM Shield-1 were fixed in 4% PFA and counterstained with Hoechst. Scale bar is 20 μm. **b** DAOY cells stably expressing NLS-AgDD-sfGFP or NLS-AgDD-NT\*-sfGFP in the absence or presence of 1 μM Shield-1 were fixed in 4% PFA and counterstained with Hoechst. Scale bar is 20 μm.

compartments of human cells. This is of particular interest for the large number of human diseases that are linked to the intracellular aggregation of amyloidogenic proteins.

We opted to use for our proof-of-principle study the regulatable aggregation AgDD domain for a number of reasons. First, aggregation-prone, disease-associated proteins are typically toxic for cells and their presence causes cellular dysfunction, often followed by cell death[4]. Hence, expression of such proteins triggers adaptive responses[34]. The regulatable nature of the AgDD domain allowed us to acutely induce aggregation by omission of the Shield-1 ligand and thereby minimizing the risk for interference of adaptive reponses[22]. Second, disease-associated, aggregation proteins are involved in various cellular processes, which may be enhanced by the presence of the aggregation-prone domain[35], and overexpression of wild-type or mutant proteins is therefore likely to alter the physiological state of the cell. We aimed to circumvent this by overexpression of the inert AgDD-sfGFP protein. Third, using an aggregation-prone domain instead of a disease-associated proteins, allowed us to use a modular approach in which the position of the anti-aggregation domain could be easily altered.

In both the cytosolic and nuclear compartments, introduction of the NT\* domain could prevent the ligand-regulatable aggregation of the reporter. This effect was independent of the position of the NT\* domain in the host protein and comparable to the solubilizing effect of the Shield-1 ligand. Whereas administration of Shield-1 resulted in enrichment of the solubilized reporter in nucleoli, this was not observed when aggregation was prevented by inclusion of the NT\* domain. Considering the pivotal role of nucleoli in the sequestration of misfolded proteins[36], this may indicate that NT\*-containing fusion has an increased structural integrity than the AgDD-sfGFP fusion in the presence of Shield-1. The position-independent effect of the NT\* domain is promising when it comes to preventing aggregation of disease-associated proteins as it suggests a plasticity that may allow construction of generic tools for targeting endogenous alleles. It is noteworthy that even though the NT\* domain prevented ligand-regulatable aggregation of the nuclear targeted reporter, it at the same time

gave rise to a small insoluble pool that was accompanied in HeLa cells by a granular nuclear distribution of the reporter protein. The reason for this phenomenon is presently unclear but appears to be a feature of the NT\* domain itself.

Long-term efforts have focused on the identification of small molecules and peptides that can prevent, inhibit or reverse protein aggregation[37]. Although there is little doubt that such drug-like molecules would be preferable from a therapeutic perspective, exploiting inhibition of protein aggregation by natural domains has some advantages that justify efforts to better understand their mode of action and applicability in the context of human diseases. Importantly, whereas compounds that modulate protein aggregation are unlikely to be specific for disease-associated proteins and may interfere at the same time with physiological processes driven by amyloidogenic conversion of proteins, the *cis*-acting nature of NT\* and other natural anti-aggregation domains will confine their effect to the host protein only while leaving other processes unaffected. This rather unique feature may be hard to accomplish with drug-like molecules but become a more realistic endeavor with increased understanding of natural domains that have evolved to control protein aggregation. With this in mind, we anticipate that our finding of the conserved anti-aggregation activity of a spider protein in human cells may stimulate further interest in these domains.

## Materials and Methods

**Constructs**. The AgDD-sfGFP construct was purchased from Addgene (#78289). All PCR amplifications were done according to the manufacturer's instructions for Phusion HF DNA polymerase (New England Biolabs, Ipswich, USA). The DD domain was PCR amplified using the primers 5'-CAG TGC TGG GAA TTC AAA TTC GGC CAC CAT GCT GG-3' (fwd) and 5'-AAG TTC TTC TCC TTT GCT GAA TTC TTC CGG TTT TAG AAG CTC CAC-3' (rev). The NT\*-tag was PCR amplified using the previously described NT\*-Aβ42 construct[14] and the primers 5'-GTC TCA TCA TTT TGG CAA AGA TGT CAC ACA CTA CAC CAT G-3' (fwd) and 5'-CAT GGT GGC CGA ATT TGA ATT CCC AGC ACT GAC-3' (rev). The PCR products were purified from a 1% agarose gel using the Qiagen gel extraction kit (Qiagen, Venlo, Netherlands) and inserted in EcoR1 digested AgDD-sfGFP using NEbuilder HiFi DNA Assembly Master Mix (New England Biolabs, Ipswich, USA). In order to clone AgDD-NT\*-sfGFP, a similar procedure was performed with the primers 5'-AAA CCG GAA GAA TTC ATG TCA CAC ACT ACA CCA

TG-3' (fwd) and 5'-AGT TCT TCT CCT TTG CTG AAG CTA GCT GAA TTC CCA GC-3' (rev). For the DD domain, the primers 5'-GTC TCA TCA TTT TGG CAA AGA ATT CGC CAC CAT GCT GG-3' (fwd) and 5'-TAG TGT GTG ACA TGA ATT CTT CCG GTT TTA GAA GCT CCA C-3' (rev) were used. The AgDD-sfGFP-NT* was originally constructed in the EGFP-N1 vector (Clontech Laboratories, Mountain View, USA) by digesting EGFP-N1 with the restriction enzymes BamHI and NotI. The AgDD-sfGFP was PCR amplified using the primers 5'-CGA CGG TAC CGC GGG CCC GGG ATC CAT GCT GGC CCT GAA GCT G-3' (fwd) and 5'-TAG TGT GTG ACA TGA ATT CGG ATC CTT TGT AGA GCT CAT CCA TG-3' (rev). NT* was amplified using the primers 5'-TGA GCT CTA CAA AGG ATC CGA ATT CAT GTC ACA CAC TAC ACC-3' (fwd) and 5'-TGA TTA TGA TCT AGA GTC GCG GCC GCT TAT GAA TTC CCA GCA CTG AC-3' (rev). The AgDD-sfGFP and NT* PCR products were inserted in the EGFP-N1 backbone according to the procedure outlined above giving rise to AgDDsfGFP-NT* (EGFP-N1). The AgDD-sfGFP-NT* open reading frame was PCR amplified from AgDDsfGFP-NT* (EGFP-N1) the using the primers 5'-AAA CAT TCT TGG ACA CAA ACT CGA GTA CAA CTT TAA CTC ACA C-3' (fwd) and 5'-TAA GCT GCA ATA AAC AAG TTA ACT TAT GAA TTC CCA GCA CTG-3' (rev). The PCR product was used to swap the insert in AgDD-sfGFP plasmid, which was excising with the restriction enzymes HpaI and XhoI, with AgDD-sfGFP-NT* using the procedure outlined above. NLS-AgDD-sfGFP was received from Add-gene (#80625). The NLS-AgDD-NT*-sfGFP was generated by PCR amplifying the open reading frame from NLS-AgDD-sfGFP (Addgene #80625) using the primers 5'-GTC TCA TCA TTT TGG CAA AGA ATT CGC CAC CAT GCC AC-3' (fwd) and 5'-TAG TGT GTG ACA TGAA TTC TTC CGG TTT TAG AAG CTC CAC-3' (rev). The PCR product was inserted with the PCR amplified NT*-tag using NEbuilder HiFi DNA Assembly Master Mix. The NT*-sfGFP was generated by assembling single strand DNA oligo 5'-GTA TGA ATG ATG TCA GTG CTG GGA ATT CAG CTA GCT TCA GCA AGG AGA AGA ACT TTT CA-3' (NT*-sfGFP) with EcoRI digested NT*-AgDD-sfGFP plasmid, using NEbuilder HiFi DNA Assembly Master Mix.

**Generation of stable cell lines**. Parental HeLa cells and DAOY cells were transfected with either AgDD-sfGFP, NT*-AGDD-sfGFP, AgDD-NT*-sfGFP or AgDD-sfGFP-NT* together with 0,2 μg PiggyBac Transposase Expression Vector using Lipofectamine 3000 (Life Technologies, Carlsbad, USA) according to manufacturer's instructions. Transfected cells were successively sorted by flow cytometry for sfGFP expression at 3 and 10 days (HeLa cells) or only at 3 days (DAOY cells) after transfection using the ARIA III FACS sorter followed by expansion of the sorted cells.

**Fluorescence microscopy**. Stable cell lines were seeded on coverslips in the presence or absence of 1 μM Shield-1 (Takara Bio, Kyoto, Japan). Cells were fixed with 4% paraformaldehyde (PFA) in phosphate buffered saline (PBS) for 15 min at 24 or 48 h postseeding. Coverslips were washed, stained with 2 μg/ml Hoechst 33342 (Thermo Fischer Scientific) in PBS for 15 min and mounted with homemade Mowiol/DABCO. Coverslips were imaged using a Zeiss LSM 880 confocal microscope with a Plan-Apo 40x/1.3 Oil objective (Zeiss, Jena, Germany) or a Zeiss LSM 710 confocal microscope equipped with a C-Apochromat 40X/1.2 water objective.

**Soluble and insoluble fractionation**. HeLa cells stably expressing AgDD-sfGFP, NT*-sfGFP, AgDD-NT*-sfGFP or AgDD-sfGFP-NT* were incubated in the absence or presence of 1 μM Shield-1 for 16-24 h. Cells were washed twice with PBS, harvested in lysis buffer A (150 mM NaCl, 50 mM Tris pH 7.5, 0.5 mM EDTA, 1% Triton X-100, 1× complete protease inhibitor cocktail (Roche, Basel, Switzerland), 10 μM MG132) and incubated on ice for 50 min. A small part of the sample was taken and mixed with 0.5 of its volume of 4x reducing NuPAGE LDS sample buffer (Invitrogen, Carlsbad, USA) and boiled ("Input"). The remaining sample was spun down at 21.130 g and 4 ºC for 30 min. The supernatant was mixed with 0.5 of its volume of 4x reducing NuPAGE LDS sample and boiled ("Soluble fraction"). The pellet was washed twice with lysis buffer and resuspended in equal volumes of lysis buffer and 4x reducing NuPAGE LDS sample buffer. These samples were boiled and sonicated ("Insoluble fraction"). Samples were run on 4–12% NuPAGE Bis-Tris gels (Invitrogen, Carlsbad, USA) with MOPS running buffer and transferred to PVDF membranes. The membranes were blocked in 5% milk dissolved in Tris-buffered saline (TBS) + 0,1% Tween and incubated first with primary antibodies against GFP (Abcam, AB290, rabbit polyclonal, 1:2500 in milk or Roche, #1814460, mouse, 1:5000 in milk), against the NT*-tag[38] (rabbit, 1:1500 in milk) and GAPDH (Abcam, ab29485, rabbit, 1:2500), followed by IRDye 800CW goat anti-mouse IgG and IRDye 680RD goat anti-rabbit IgG secondary antibodies (both 1:5000) (LI-COR, Lincoln, USA) and imaged with the Odyssey infrared imaging system (LI-COR, Lincoln, USA). The intensity of the GFP bands in the "Soluble fraction" and "Insoluble fraction" of lysates collected in three independent experiments - performed on separate days - of the soluble and insoluble fractions were quantified using ImageJ.

**Statistics and reproducibility**. Quantifications of western blots were obtained from three independent experiments. All experiments have been checked for reproducibility.

**Reporting summary**. Further information on research design is available in the Nature Research Reporting Summary linked to this article.

## Data availability

All original data can be obtained from the corresponding author upon request. Uncropped blots are shown in Supplementary Fig. 2. The original data of the graphs is shown in Supplementary Data 1.

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

## Acknowledgements

We thank the members of the Dantuma lab, Dr. Henrik Biverstal, Dr. Per Nilsson and Dr. Makoto Shimozawa (Department of Neurobiology, Care Sciences and Society, Karolinska Institutet) for helpful input. This work was supported by the Swedish Research Council (N.P.D. 2016-02479, J.J. 2020-02434), the Swedish Cancer Society (N.P.D. CAN 2018/693), CIMED (J.J.) and StratNeuro (N.P.D., J.J.). S.X. was supported by a scholarship from Chinese Scholarship Council (CSC). A.K.S. and M.E.G. were supported by research fellowships from the Deutsche Forschungsgemeinschaft (DFG) (SHE 2079/1-1; GI-1329/1-1).

## Author contributions

A.K.S., S.X. performed all experiments except for the flow cytometry; M.G., S.X. performed flow cytometry; A.K.S., S.X., J.V. performed the microscopy experiment shown in Figs. 2 and 3; A.K.S., J.J., N.P.D. wrote the manuscript; J.J., N.P.D. coordinated the project; all authors edited and approved the final manuscript.

## Funding

## Competing interests

The authors declare no competing interests.
