## [Peer Review File · Communications Biology]

Reviewers' comments:

Reviewer #1 (Remarks to the Author):

This authors have investigated the effect of the engineered spidroin NT* domain on induced aggregation within human cells. The domain has previously been developed and applied as a solubility and expression aid for partner proteins. Here they have combined the NT* domain with the small molecule Shield-1 FKBP ligand and have determined the effect of the NT* domain on modulation of aggregation, in the cytosol and nucleus of HeLa cells.

This is a useful extension of previous work with the NT* domain and potentially opens up further avenues for use of the domain in biotechnology applications and as a tool for investigating the cellular impact of aggregation and rescue, with a variety of client proteins. It is likely to be of interest to many in the field.

The experimental results are of high quality and clearly described but there are some questions unanswered, regarding the generalisability of the system to other client proteins and impact of nuclear localisation. The inclusion of a disease-associated, aggregation-prone client protein for comparison with the inducible aggregation here, would greatly add to the impact of this work.

Points to be addressed:

The authors should include a brief explanation of the nature of the two-residue modification that generates the NT* domain – i.e. explain specifics of residue charge switch. Likewise, brief description of the AgDD is important, nature of FKBP domain and Shield-1, should be included in the Introduction.

Authors should comment on the effect of position of the NT* domain on the levels of protein expression achieved.

Line 129 Explain what is meant by “homogeneous expression of the transgene”. How is this detected by flow cytometry? Is this referring to similar levels with different constructs, or cellular distribution, or lack of puncta?

The images in Fig 2E show that the inclusion of the NT* domain leads to uniform, cytosolic distribution and no obvious GFP-positive puncta. However, Fig 2F shows that addition of Shield ligand leads to an apparently even distribution across nucleus and cytoplasm. Is there other evidence in the literature that the activity or response of the AgDD system is affected by nuclear or cytosolic localisation? The authors should include an NT*-GFP constructs as additional control to discriminate between effects of the ligand and NT* domain.

Additionally, the results presented Fig 3, where the NT* domain only provides ~50% rescue from aggregation in the absence of Shield-1 and inhibits rescue by Shield-1, require further investigation. Do the authors have other evidence that would report on the effect of an NLS on NT* activity and function? The authors should include NLS-NT*-GFP constructs or other data indicating that the NLS does not affect NT*. Have the authors tested the effect of position of the NT* in these experiments, e.g. in NLS-AgDD-sfGFP-NT*?

Please explain what is meant by “three independent experiments of the soluble and insoluble fractions”? Were fractions prepared from three independent biological replicates?

Reviewer #2 (Remarks to the Author):

This work presented a study using spider silk-derived soluble domain to preventing protein aggregation in the human cell. Though the overall quality of the study is above average, some serious concerns have to be considered.

1. More background or discussion on using such a strategy to prevent protein aggregation, particularly disease-related protein aggregation, is needed. In other words, the importance of this study has to be adequately highlighted.
2. What is the benefit of using such a strategy over small molecules, especially from the clinical translation point of view.
3. In addition, the study used a reporter protein whose aggregation is ligand-regulatable. Why not using directly disease-related proteins.
4. The mechanism for the independence of the position of the NT* domain in the host protein is unclear. This should be studied. Are there any general rules for designing such domains with similar functions to spider silk-derived NT*. Is spider silk-derived NT* unique?
5. In the title, "solubility domain" should be "soluble domain". The title is too big and should be tuned down since the whole study was done on a reporter protein.

Reviewer #3 (Remarks to the Author):

In the manuscript "A spider silk-derived solubility domain inhibits nuclear and cytosolic protein aggregation in human cells", Schellhaus et al., studied the potential of a recombinant spider silk-derived NT* domain in preventing protein aggregation in HeLa cells. In cells expressing recombinant NT* domain in fusion with an aggregation reporter, the soluble pool of the reporter increased and the propensity to aggregate was independent of the relative position of the NT* domain with respect to the aggregation reporter. The authors went on to conclude that the spider silk-derived NT* domain has a generic anti-aggregation activity, independent of position or subcellular location, that is also active in human cells and propose that the NT* domain can potentially be exploited in controlling protein aggregation of disease-associated proteins.

The study is interesting from a basic science perspective, builds over their previous work on the anti-aggregation potential of NT*, and together with earlier work, builds a strong rationale for testing the functions of such recombinant proteins in vivo. However, the current manuscript suffers from some issues, as outlined below, which renders it unsuitable for publication in its present form. This reviewer realizes the potential of the study and is willing to re-review a significantly revised manuscript for consideration in *Communications Biology*.

Major revision:

The physiological relevance of protein misfolding and aggregation has been studied most widely in the contexts of neurodegenerative disorders. Even in the references cited by the authors to introduce the generality of protein aggregation, the major focus has been the proteins related to amyloid or neurodegenerative disorders. In this context, this reviewer would feel more confident of the anti-aggregation potential of NT* in cells that are more relevant to such conditions than HeLa cells e.g., SH-SY5Y cells or any other relevant model. A didactic review in this regard was published in 2020 by Slanzi et al. (doi: 10.3389/fcell.2020.00328). This is because the secretory system for cellular trafficking of aggregation-prone proteins are differentially adapted in those specialized cell types than HeLa cells. The authors may use one such model to reproduce a subset of the results from HeLa cells to significantly improve the scientific appeal of the manuscript.

Minor revision:

1. The authors should revise the statement (line 119) that states " Administration of Shield1 did not further increase the solubility of AgDD-NT*-sfGFP suggesting that protein aggregation was efficiently prevented by insertion of the NT* domain." Fig.1c shows a quantifiable difference between the S and P panels in the absence or presence of S1.
2. All western blot experiments have been done on transiently transfected cells, whereas flow and microscopy on stable cell lines. It is important to repeat a subset of western blots on stable cell lines or provide reliable quality control data for transfection efficiency.

Rebuttal

Authors: We would like to thank the reviewers for their constructive comments. The revised manuscript includes new experimental data and textual edits to address the concerns of the reviewers.

Reviewer 1

This authors have investigated the effect of the engineered spidroin NT* domain on induced aggregation within human cells. The domain has previously been developed and applied as a solubility and expression aid for partner proteins. Here they have combined the NT* domain with the small molecule Shield-1 FKBP ligand and have determined the effect of the NT* domain on modulation of aggregation, in the cytosol and nucleus of HeLa cells.

This is a useful extension of previous work with the NT* domain and potentially opens up further avenues for use of the domain in biotechnology applications and as a tool for investigating the cellular impact of aggregation and rescue, with a variety of client proteins. It is likely to be of interest to many in the field.

The experimental results are of high quality and clearly described but there are some questions unanswered, regarding the generalisability of the system to other client proteins and impact of nuclear localisation. The inclusion of a disease-associated, aggregation-prone client protein for comparison with the inducible aggregation here, would greatly add to the impact of this work.

Authors: We are pleased that the reviewer appreciates the importance of our findings for the field. Below we discuss the concerns raised by the reviewer and how they have been addressed in the revised manuscript.

Points to be addressed:

The authors should include a brief explanation of the nature of the two-residue modification that generates the NT* domain – i.e. explain specifics of residue charge switch. Likewise, brief description of the AgDD is important, nature of FKBP domain and Shield-1, should be included in the Introduction.

Authors: We have included in the Introduction section a better explanation of the NT* and AgDD domain and included additional citations. See lines 70-77 (NT*) and lines 87-89 (AgDD).

Authors should comment on the effect of position of the NT* domain on the levels of protein expression achieved.

Authors: We included at the point where we present the FACS analysis a comment about the relative steady-state levels of the different reporter proteins in the stable cell lines. See lines 133-138.

Line 129 Explain what is meant by “homogeneous expression of the transgene”. How is this detected

by flow cytometry? Is this referring to similar levels with different constructs, or cellular distribution, or lack of puncta?

Authors: We agree that this statement was unclear but were referring to the fact that the expression levels within each clone were similar as we obtained a Gaussian distribution indicative for a single population. We have replaced this statement with a more precise comment about the expression levels. See lines 133-138.

The images in Fig 2E show that the inclusion of the NT* domain leads to uniform, cytosolic distribution and no obvious GFP-positive puncta. However, Fig 2F shows that addition of Shield ligand leads to an apparently even distribution across nucleus and cytoplasm. Is there other evidence in the literature that the activity or response of the AgDD system is affected by nuclear or cytosolic localisation? The authors should include an NT*-GFP constructs as additional control to discriminate between effects of the ligand and NT* domain.

Authors: As the reviewer proposed, we have generated and tested an NT*-sfGFP construct. The NT*-sfGFP is equally distributed between the cytosolic and nuclear compartments in the absence or presence of ligand (Suppl. Fig. 1). This implies that the cytosolic localization in the absence of ligand of constructs containing both the NT* and AgDD domain can be attributed to the AgDD domain. This is also supported by the observation that the AgDD-sfGFP forms cytosolic aggregates but localizes to the nucleus in the presence of ligand, implying that the NT* domain is not required for this phenomenon. To the best of our knowledge, it has not been reported before that the Shield1 ligand affects the localization of AgDD fusion proteins. The AgDD fusions with the NT* domain may have made this feature more apparent as it keeps the fusion in a soluble state also in the absence of ligand. We discuss this in the Result section of the paper. See lines 159-162.

Additionally, the results presented Fig 3, where the NT* domain only provides ~50% rescue from aggregation in the absence of Shield-1 and inhibits rescue by Shield-1, require further investigation. Do the authors have other evidence that would report on the effect of an NLS on NT* activity and function? The authors should include NLS-NT*-GFP constructs or other data indicating that the NLS does not affect NT*.

Authors: Since the NT* does not have any effect in the context of an NLS-NT*-GFP fusion (as it is not aggregation prone), it would not be possible to test with the proposed construct whether the NLS affect the NT* domain. We do, however, show that the NT* domain prevents the aggregation of the NLS-AgDD-sfGFP-NT* in HeLa and DAOY cells, which demonstrates that the NLS domain does not interfere with this property of the NT* domain. Vice versa the NT* domain does not affect the NLS domain as this fusion still properly localizes to the nuclei in both cell lines.

Have the authors tested the effect of position of the NT* in these experiments, e.g. in NLS-AgDD-sfGFP-NT*?

Authors: We have not tested the NLS-AgDD-sfGFP-NT* domain but we have shown that 1) the NT* domain does not seem to interfere with the nuclear localization and 2) we have tested the functionality of the NT* domain in various positions for the AgDD-sfGFP fusions.

Please explain what is meant by “three independent experiments of the soluble and insoluble fractions”? Were fractions prepared from three independent biological replicates?

Authors: With three independent experiments, we meant that the entire procedure (seeding, transfection, harvesting) has been performed on separate occasions. We were not referring to replicate samples. We did not have the feeling that this description was ambiguous and open for different interpretations but tried to outline it even more explicit in the revision. See lines 363-365.

Reviewer #2

This work presented a study using spider silk-derived soluble domain to preventing protein aggregation in the human cell. Though the overall quality of the study is above average, some serious concerns have to be considered.

Authors: We are pleased to hear that the reviewer felt that the quality of the study is above average. We would like to thank the reviewer for helpful suggestions to strengthen the manuscript. Below, we discuss how the suggestions have been addressed.

1. More background or discussion on using such a strategy to prevent protein aggregation, particularly disease-related protein aggregation, is needed. In other word, the importance of this study has to be adequately highlighted.

Authors: In the revision, we open the Discussion section with highlighting the importance to develop strategies for preventing protein aggregation. See lines 208-219.

2. What is the benefit of using such a strategy over small molecules, especially from the clinical translation point of view.

Authors: From a clinical point of view, small molecules will be desirable for preventing protein aggregation by therapeutic intervention. However, we believe that understanding the mode of action of natural anti-aggregation domains may aid the development of small molecules and/or therapeutic strategies that prevent protein aggregation. This is explained in lines 262-276.

3. In addition, the study used a reporter protein whose aggregation is ligand-regulatable. Why not using directly disease-related proteins.

Authors: There are various reasons why we opted for using an engineered, ligand-regulatable aggregation domain. We do appreciate that the next step will be to evaluate the effectiveness of the NT* domain in the context of a disease-related protein but feel that this is a topic for further research and lies outside the scope of the present paper.

4. The mechanism for the independence of the position of the NT* domain in the host protein is unclear. This should be studied. Are there any general rules for designing such domains with similar functions to spider silk-derived NT*. Is spider silk-derived NT* unique?

Authors: Solubility tags are as a rule attached N-terminally of the target proteins (see eg Esposito, D. & Chatterjee, D. K. 2006. Enhancement of soluble protein expression through the use of fusion tags. *Curr. Opin. Biotechnol.* 17, 353–358). This location ensures that the solubility tag is already present when the target protein is being produced and allows proteolytic removal of the tag without leaving protease recognition motifs in the target protein. Unpublished data from our lab show that recombinant production in bacteria of amyloid-beta peptide, islet amyloid polypeptide and nerve growth factor as target proteins give equally much soluble fusion protein with N- or C-terminally located NT* and similar data are published for other tags.

5. In the title, “solubility domain” should be “soluble domain”. The title is too big and should be tuned down since the whole study was done on a reporter protein.

Authors: “Solubility domain” is a term that is general used for domains that increase the solubility of a protein. We feel that “soluble domain” can be misunderstood as it may be interpreted as a reference to the fact that the domain itself is soluble without indicating whether it affects the solubility of the entire protein.

Reviewer #3 (Remarks to the Author):

In the manuscript "A spider silk-derived solubility domain inhibits nuclear and cytosolic protein aggregation in human cells", Schellhaus et al., studied the potential of a recombinant spideroin-based NT* domain in preventing protein aggregation in HeLa cells. In cells expressing recombinant NT* domain in fusion with an aggregation reporter, the soluble pool of the reporter increased and the propensity to aggregate was independent of the relative position of the NT* domain with respect to the aggregation reporter. The authors went on to conclude that the spideroin-derived NT* domain has a generic anti-aggregation activity, independent of position or subcellular location, that is also active in human cells and propose that the NT* domain can potentially be exploited in controlling protein aggregation of disease-associated proteins.

The study is interesting from a basic science perspective, builds over their previous work on the anti-aggregation potential of NT*, and together with earlier work, builds a strong rationale for testing the functions of such recombinant proteins in vivo. However, the current manuscript suffers from some issues, as outlined below, which renders it unsuitable for publication in its present form. This reviewer realizes the potential of the study and is willing to re-review a significantly revised manuscript for consideration in *Communications Biology*.

Authors: We are pleased that the reviewer finds the study interesting and having potential. Below we outline the additional experimentation that has been done to address the concerns of the reviewer.

Major revision:

The physiological relevance of protein misfolding and aggregation has been studied most widely in the contexts of neurodegenerative disorders. Even in the references cited by the authors to introduce the generality of protein aggregation, the major focus has been the proteins related to

amyloid or neurodegenerative disorders. In this context, this reviewer would feel more confident of the anti-aggregation potential of NT* in cells that are more relevant to such conditions than HeLa cells e.g., SH-SY5Y cells or any other relevant model. A didactic review in this regard was published in 2020 by Slanzi et al. (doi: 10.3389/fcell.2020.00328). This is because the secretory system for cellular trafficking of aggregation-prone proteins are differentially adapted in those specialized cell types than HeLa cells. The authors may use one such model to reproduce a subset of the results from HeLa cells to significantly improve the scientific appeal of the manuscript.

Authors: We agree with the reviewer that repetition of the key findings in cell line that has more relevance to neurodegenerative disease can strengthen the study. Therefore, we studied the effect of the NT* in the human medulloblastoma cell line DAOY. Initially attempts to generate stable cell lines with SH-SY5Y cells were less successful (low expression levels, large fraction of stable transfected cells was not expressing the fusion). The new data show that the NT* domain also prevented aggregation of AgDD-sfGFP in these cells independent of position of the NT* domain in the fusion protein and independent of the subcellular localization. We have also included a citation to the publication by Slanzi and co-workers and explain the importance of performing these experiments in neuron-like cell lines. See new Figure 4 and lines 190-206.

Minor revision:

1. The authors should revise the statement (line 119) that states " Administration of Shield1 did not further increase the solubility of AgDD-NT*-sfGFP suggesting that protein aggregation was efficiently prevented by insertion of the NT* domain." Fig.1c shows a quantifiable difference between the S and P panels in the absence or presence of S1.

Authors: We have removed the statement "Administration of Shield1 did not further increase the solubility of AgDD-NT*-sfGFP".

2. All western blot experiments have been done on transiently transfected cells, whereas flow and microscopy on stable cell lines. It is important to repeat a subset of western blots on stable cell lines or provide reliable quality control data for transfection efficiency.

Authors: We believe that this is based on a misunderstanding and apologize if our description has been unclear. The microscopy, flow cytometry and western blots that are shown in Fig. 2 are all performed with stable cell lines.

REVIEWERS' COMMENTS:

Reviewer #1 (Remarks to the Author):

The authors have addressed the issues raised and the additional experimental work and clarifications improve the manuscript.

The meaning of the additional text starting on line 251 is not clear ...Perhaps the authors mean the protein is more folded or correctly folded or has increased structural integrity? This should be rewritten.

"Considering the pivotal role of nucleoli in the sequestration of misfolded proteins³⁶, this may indicate that NT*-containing fusion are structurally more integer than the AgDD-sfGFP fusion in the presence of Shield1."

Reviewer #2 (Remarks to the Author):

The authors have addressed all my comments well and the manuscript merits the publication in the journal.

Reviewer #3 (Remarks to the Author):

The authors have addressed my concerns. This manuscript is acceptable for publication after the two minor corrections below:

1. delete the word 'typically' in line 195.
2. Add scale bar in Fig. 4B.